biochemistry/chemical ecology/environmental chemistry

re-release, *Chlorella vulgaris*, ferric phosphate, biomass, phosphorus removal, eutrophication

**Author for correspondence:**
Yongcan Chen
e-mail: swust_chen@189.cn

# Effects of *Chlorella vulgaris* on phosphorus release from ferric phosphate sediment by consecutive cultivations

Lile He[1], Yongcan Chen[1], Shu Chen[1], Xuefei Wu[1] and Jing Liu[2]

[1]Key Laboratory of Solid Waste Treatment and Resource Recycle of Ministry of Education, Southwest University of Science and Technology, Mianyang 621010, People's Republic of China
[2]College of Resources and Environment, Southwest University, Chongqing 400715, People's Republic of China

ⓘ YC, 0000-0002-1608-1269

Iron phosphate (Fe–P) is a main phosphorus storage form, especially in phosphorus-polluted environments. The re-release of Fe–P is a problematic result during microalgal remediation. In this study, pre-incubated *Chlorella vulgaris* was cultured in a BG-11 culture medium with different amounts of Fe–P. The effects of Fe–P re-release on biomass, flocculation and removal of $PO_4^{3-}$ were investigated. The results indicated that *C. vulgaris* can promote the dissolution and release of Fe–P when the pH is 7, and the amount of Fe–P ($\Delta Q$) released in 200 ml water reaches 0.055–0.45 mg d$^{-1}$ under a *C. vulgaris* concentration of $5.6 \times 10^5$–$8 \times 10^5$ cells ml$^{-1}$. The growth of *C. vulgaris* was inhibited because of the flocculation behaviour of $Fe^{3+}$ in the release stage, which is associated with a specific growth rate of 0.3–0.4 d$^{-1}$ and a phosphorus removal rate below 30%. However, this process, in the long term, indicates a favourable transformation in which Fe–P becomes bioavailable under the action of *C. vulgaris*. Microalgae outbreaks may be triggered by persistent interactions between Fe–P and *C. vulgaris*. This study provides an important reference for the application of *C. vulgaris* in a Fe–P-rich environment.

## 1. Introduction

Domestic sewage, agricultural effluent and industrial wastewater containing large amounts of phosphorus (P) discharged into river water significantly promote the eutrophication of water bodies, which causes multiple unwanted environmental impacts [1,2]. *Chlorella vulgaris* populations grow by using inorganic

This article has been edited by the Royal Society of Chemistry, including the commissioning, peer review process and editorial aspects up to the point of acceptance.

nutrients, including nitrogen (N) and P, which means it has a high potential for removing inorganic nutrients from the water. Therefore, *C. vulgaris* has been used for the treatment of eutrophic water bodies [3]. From the perspective of biomaterials, *C. vulgaris* shows a high metal binding capacity because of the presence of polysaccharides and various functional groups of proteins or lipids on the cell wall surface (such as amino, hydroxyl, carboxyl and sulfate) that can serve as binding sites for metal ions [4]. Currently, *C. vulgaris* has been widely used in wastewater treatment (domestic sewage, industrial wastewater and municipal wastewater) based on its optimal adsorption or absorption capacity for N, P and metal cations (such as $Fe^{3+}$, $Cr^{5+}$, $Ni^{2+}$, $Zn^{2+}$, $Cu^{2+}$ and $Pb^{2+}$) [5–10].

P in rivers and lakes is mainly present as $PO_4^{3-}$ [11], which can readily combine with metal cations (such as $Fe^{3+}$, $Ca^{2+}$ and $Al^{3+}$) to form a precipitate mainly sequestered in the sediments [12]. Iron phosphate (Fe–P) is a combination of $PO_4^{3-}$ and $Fe^{3+}$ and is considered an active compound with a high release rate [13]. It has important effects on the overlying water because of its frequent release and associated periodic outbreaks of microalgae [14]. The release of $PO_4^{3-}$ and $Fe^{3+}$ from Fe–P could cause persistent pollution of the water column, and this could occur via the actions of microorganisms or by human disturbance [15,16]. Zhu *et al.* [17] studied the internal nutrient release response to an algal bloom decay in Taihu Lake in China, showing that sediment from the river mouth releases P to the water in the presence of blooms.

The mechanism of Fe–P release under the action of *C. vulgaris* has not been clarified. Several mechanisms may explain the release of Fe–P. First, reductive dissolution is caused by the respiration of *C. vulgaris* under anoxic conditions at the sediment–water interface. Second, the competitive behaviour of $OH^-$ and $PO_4^{3-}$. Third, the reverse reaction of $Fe^{3+}$ hydrolysis is enhanced by a weakly acidic environment [18,19].

Released $Fe^{3+}$ from Fe–P can inhibit the growth of *C. vulgaris* and even exhibit a toxic effect, which is not a favourable result when treating eutrophic wastewater [20,21]. Periodical algal blooms are induced by the release of P from P-containing sediments [22,23]. In addition, $Fe^{3+}$ has a strong flocculation effect on *C. vulgaris* because the surface of *C. vulgaris* is negatively charged and evenly suspended in liquid [24,25]. Therefore, chain reactions involving biomass production, pH fluctuation, flocculation, co-precipitation and removal of P can be triggered through the release dynamics of Fe–P. However, these processes have not been used to evaluate the advantages and disadvantages of *C. vulgaris*.

Phosphorus-contaminated water bodies can be mediated by *C. vulgaris* in a Fe–P-rich environment [26–29]. To simulate actual environmental conditions, using different amounts of $FeCl_3$ and high-content $K_2HPO_4$ to represent Fe–P flocculated sediments, *C. vulgaris* was cultured in a BG-11 medium during a continuous three-cycle culture. The effects of $Fe^{3+}/PO_4^{3-}$ release from Fe–P on biomass of *C. vulgaris*, $PO_4^{3-}$ removal and flocculation were investigated. As such, this study evaluated the net environmental effects of *C. vulgaris* in Fe–P-rich river sediments providing a model for similar studies in other water bodies.

# 2. Material and methods

## 2.1. *Chlorella vulgaris* strain and culture medium

The *C. vulgaris* strain (serial number: FACHB-8) was obtained from the Freshwater Algae Culture Collection at the Institute of Hydrobiology, China. The standard composition of the BG-11 medium was as follows: $NaNO_3$ (10 ml $l^{-1}$ in 15 g 100 $ml^{-1}$); $K_2HPO_4$ (10 ml $l^{-1}$ in 5 g $l^{-1}$); $MgSO_4 \cdot 7H_2O$ (10 ml $l^{-1}$ in 7.5 g $l^{-1}$); $CaCl_2 \cdot 2H_2O$ (10 ml $l^{-1}$ in 3.6 g $l^{-1}$); citric acid (10 ml $l^{-1}$ in 0.6 g $l^{-1}$); ferric ammonium citrate (10 ml $l^{-1}$ in 0.6 g $l^{-1}$); $EDTANa_2$ (10 ml $l^{-1}$ in 0.1 g $l^{-1}$); $Na_2CO_3$ (10 ml $l^{-1}$ in 2 g $l^{-1}$); and $A_5$ (1 ml $l^{-1}$). The composition of $A_5$ metal solution was as follows: $H_3BO_3$ (2.86 g $l^{-1}$); $MnCl_2 \cdot 4H_2O$ (1.86 g $l^{-1}$); $ZnSO_4 \cdot 7H_2O$ (0.22 g $l^{-1}$); $Na_2MoO_4 \cdot 2H_2O$ (0.39 g $l^{-1}$); $CuSO_4 \cdot 5H_2O$ (0.08 g $l^{-1}$); and $Co (NO_3)_2 \cdot 6H_2O$ (0.05 g $l^{-1}$).

## 2.2. Experimental solutions and instruments

All chemicals were of analytical reagent grade and were used without further purification. All solutions and algal suspensions were prepared using Milli-Q water. The $Fe^{3+}$ and $PO_4^{3-}$ solutions in the experiment were prepared from analytically pure $FeCl_3 \cdot 6H_2O$ and $K_2HPO_4$, respectively. Specifically, 1 g $l^{-1}$ $Fe^{3+}$ and 4 g $l^{-1}$ $PO_4^{3-}$ stock solutions were prepared from $FeCl_3 \cdot 6H_2O$ and $K_2HPO_4$,

**Table 1.** The quantification of Fe–P and the ratio of $PO_4^{3-}$ and $Fe^{3+}$.

| conditions | control group | experimental groups | | | | | |
| --- | --- | --- | --- | --- | --- | --- | --- |
| P (mg l$^{-1}$) | 60 | 60 | 60 | 60 | 60 | 60 | 60 |
| Fe$^{3+}$ (mg l$^{-1}$) | non-ferrous | 0.1 | 1 | 3 | 5 | 10 | 25 |
| Fe–P (mg) | 0 | 0.02 | 0.2 | 0.6 | 1 | 2 | 5 |

respectively. $Fe^{3+}$ and $PO_4^{3-}$ solution used in the experiments were obtained by stock solution dilution. The pH of the solution (final pH = 7) was adjusted using 1 mol NaOH and HCl.

We used a constant temperature light incubator (INFORS HT Multitron, Switzerland), mounted bio-optical microscope (DM2000, Germany), chlorophyll fluorometer (Aquafluor805186, America), centrifuge (TGL-16G, Japan), pH meter (PB-10, Sidoli Scientific Instrument Co. Ltd), high-temperature autoclave (MLS-3780) and UV spectrophotometer (Evolution 300, Shanghai Yuanqing Instrument Co. Ltd).

## 2.3. Experimental design

Many studies have shown that the release of P is related to the speciation of P in sediments [30]. Fe–P is a soluble reactive P at the sediment/water interface; the release of P from Fe–P can be caused by the coupling mechanisms of Fe–P [18]. To simulate the formation of Fe–P in the water body as much as possible, high-content $K_2HPO_4$ and gradient concentrations of $FeCl_3$ were used to synthesize Fe–P. The chemical reaction equation is given by equation (2.1).

$$FeCl_3 + K_2HPO_4 = FePO_4 \downarrow + HCl + 2KCl. \qquad (2.1)$$

The preliminary preparation was as follows: 250 ml Erlenmeyer flasks (14 groups) were set up as control and experimental groups (seven groups) with parallel samples (seven groups), and 200 ml of BG-11 standard medium (with ferric ammonium citrate and $K_2HPO_4$ removed) were decanted to an Erlenmeyer flask after measuring the initial biomass. Ferric phosphate precipitate was formed by adding a fixed amount of $FeCl_3$ (0.1, 1, 3, 5, 10 and 25 mg l$^{-1}$) in a medium containing 60 mg l$^{-1}$ $K_2HPO_4$. The quantification of $Fe^{3+}$, $PO_4^{3-}$ and the final Fe–P solid phase are given in table 1.

The specific steps were as follows. *Chlorella vulgaris* was pre-cultured in a culture consisting of saturated iron ammonium and $K_2HPO_4$; 50 ml *C. vulgaris* in a stationary phase was centrifuged for 10 min (4°C, 1500 r.p.m.) after a week of cultivation. The cell block was resuspended in a conical flask with a fresh culture solution to a final volume of 200 ml (including 60 mg l$^{-1}$ $K_2HPO_4$ solution). The pH of the mixture was adjusted to 7 with a solution of NaOH or HCl 1 M after a fixed amount of $Fe^{3+}$ was added. All cultures were maintained in a constant-temperature light incubator at 25°C, and warm light of 2000 lux was continuously supplied by LED lamps with a day/night of 12 h/12 h. The conical flask was shaken three times a day. The experiment was conducted for a month and included four logarithmic periods of *C. vulgaris*. The data were divided into three cycles based on the experimental state. These consistency conditions constrain the fact that we did not conduct an interaction study between the effects of the factors (temperature, hydrodynamic flocculation conditions and mixing time).

## 2.4. Determination of biomass and specific growth rate

Cell counts were performed using digital images obtained through a mounted bio-optical microscope (40×) and a chlorophyll fluorometer (680 nm). The results were then transformed into the number of cells (cell ml$^{-1}$) based on the linear relationship between the number of cells and the chlorophyll level. The linear relationship is expressed as follows:

$$Biomass\,(cell\ ml^{-1}) = 4.17 \times 10^5 \times OD_{680}. \qquad (2.2)$$

The specific growth rate of *C. vulgaris* was calculated using biomass and time [15]. The equation is as follows:

$$Specific\ growth\ rate\ of\ microalgae = \left(\frac{LnX_2 - LnX_1}{T_2 - T_1}\right), \qquad (2.3)$$

where $X$ represents biomass concentrations on day T, and 1 and 2 represent the initial and final points, respectively.

## 2.5. Determination of flocculation efficiency

The flocculation efficiency was calculated using equation (2.4) [24].

$$\text{Flocculation efficiency } (\%) = \left( 1 - \left( \frac{OD_{680 \text{ Supernatant}}}{OD_{680 \text{ Initial sample}}} \right) \right) \times 100, \tag{2.4}$$

where $OD_{680 \text{ Supernatant}}$ and $OD_{680 \text{ Initial sample}}$ indicate the supernatant and initial sample, respectively. Both sets of parallel samples were tested twice.

## 2.6. Determination of removal efficiency of $PO_4^{3-}$

The removal efficiency of $PO_4^{3-}$ was calculated to compare the effect of the release of $Fe^{3+}$ on the removal of $PO_4^{3-}$ [24]. Equation (2.5) is as follows:

$$\text{Removal efficiency of } PO_4^{3-} (\%) = \left( \frac{Y_2 - Y_1}{Y_2} \right) \times 100, \tag{2.5}$$

where $Y$ represents the residual concentration of $PO_4^{3-}$ in solution, and 1 and 2 represent the initial and ending points, respectively.

## 2.7. Determination of residual concentrations of $PO_4^{3-}$ and $Fe^{3+}$ in solution

$PO_4^{3-}$ in the solution was determined by ammonium molybdate spectrophotometry. The o-phenanthroline method was used to determine the concentration of $Fe^{3+}$ in the solution. The applicable scope of the standard for $PO_4^{3-}$ and $Fe^{3+}$ determination was 0.0–50 and 0.02–20 mg $l^{-1}$, respectively. Samples were filtered (microporous membrane of 0.45 um, Jun Yuan experiment equipment, China) to detect the residual concentrations of $PO_4^{3-}$ and $Fe^{3+}$ and determined using a UV spectrophotometer (Evolution 300) at wavelengths of 710 and 510 nm, respectively. Standard curves were established using $K_2HPO_4$ and $Fe_2(SO_4)_3$. Samples whose solution concentrations exceeded the standard line were tested after dilution by a certain multiple. Specific analytical methods are described in the American Public Health Association Standard Methods.

## 2.8. Statistical analysis

Statistical analysis was run (10 times in total) using Origin 8.0, based on the total sample count (14 groups) and several iterations (two repetitions). Table 2 shows the values of F and P in the one-way analysis of variance (ANOVA). ANOVA of repeated experiments confirmed significance at the level of 0.05.

# 3. Results and discussion

## 3.1. The release of $Fe^{3+}$ and $PO_4^{3-}$

Figure 1 shows the change in $Fe^{3+}$ in solution. The variation in $Fe^{3+}$ in the three cycles exhibited obvious fluctuations. In the first period, the solution was iron-free for 0.02 and 0.2 mg Fe–P, which may be attributed to the adsorption of trace dissolved $Fe^{3+}$ by *C. vulgaris*. In the 0.6, 1, 2 and 5 mg Fe–P groups, the concentration of $Fe^{3+}$ in the solution showed an overall upward trend with an increasing amount of Fe–P, and the maximum value is close to the amount of $Fe^{3+}$ added. An increase in the $Fe^{3+}$ concentration indicates $Fe^{3+}$ re-release from Fe–P under the activity of *C. vulgaris*.

The proliferation of *C. vulgaris* can be inhibited by pretreatment with high concentrations of $PO_4^{3-}$. $Fe^{3+}$ and $PO_4^{3-}$ are adsorbed or absorbed to saturation in the original algal cells, which may have led to an evident change in $Fe^{3+}$ in the solution. Therefore, we divided the one-month experiment into three cycles based on the Fe–P re-release process. The first cycle is mainly the phase of Fe–P

**Table 2.** The value of *F* and *p* at 0.05 level in the one-way ANOVA.

| results | time (days) | | | | | | | | | |
|---|---|---|---|---|---|---|---|---|---|---|
| | 1 d | 4 d | 7 d | 10 d | 14 d | 18 d | 22 d | 24 d | 28 d | 31 d |
| *F*-value | 7.83865 | 8.96153 | 4.19505 | 6.61307 | 8.85891 | 11.9461 | 16.48771 | 38.14244 | 14.13335 | 14.83036 |
| *p*-value | 0.02194 | 0.00532 | 0.06357 | 0.01736 | 0.00551 | 0.00535 | 0.00129 | 0.00004 | 0.00134 | 0.00115 |

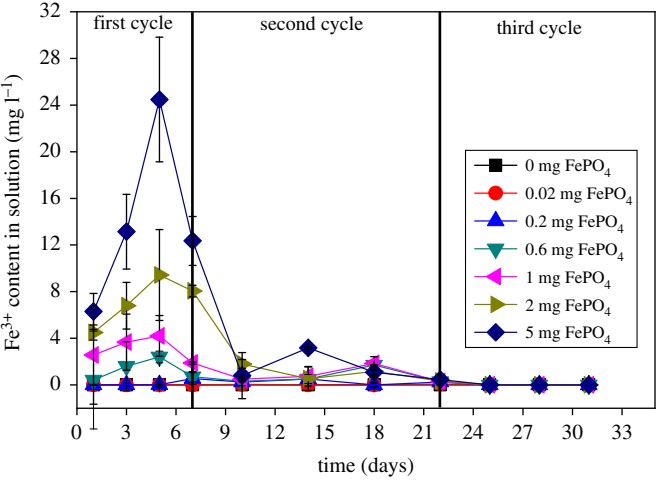

**Figure 1.** Changes of $Fe^{3+}$ concentration in the growth of *C. vulgaris*. Determination of the cycle comes from the status of $Fe^{3+}$: the first cycle is the release phase of $Fe^{3+}$, the second cycle becomes the adsorption phase of $Fe^{3+}$ in the two logarithmic growth phases of *C. vulgaris* and there is no $Fe^{3+}$ in the third cycle. Data represent the mean and s.d. of two independent experiments.

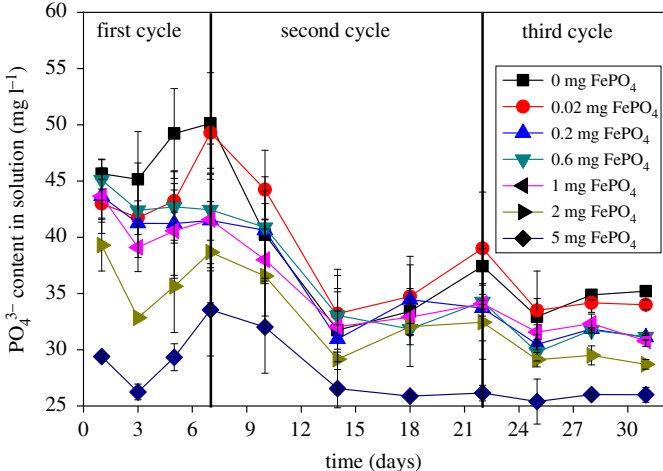

**Figure 2.** Changes of $PO_4^{3-}$ concentration in the proliferation of *C. vulgaris*. The concentration of $PO_4^{3-}$ in the solution is relatively stable with an increasing amount of $FePO_4$ after the 14th day. Data represent the mean and range of two independent experiments.

release, and the release of $Fe^{3+}$ is much greater than adsorption based on the inhibition by the high-level P. In the second cycle, the concentration of $Fe^{3+}$ continuously decreased in the groups of 0.6, 1, 2 and 5 mg Fe–P and approached 0, which may be ascribed to the adsorption and flocculation effects of *C. vulgaris*. Thus, this stage is the adsorption of *C. vulgaris*. The third cycle became the self-development stage of *C. vulgaris* because $Fe^{3+}$ was completely adsorbed. The results showed that Fe–P can be released under the action of *C. vulgaris*.

As an internal P loading, Fe–P makes a major contribution to the P content in lakes and reservoirs [31]. $PO_4^{3-}$ is also released when Fe–P is dissolved. Figure 2 shows the change in the concentration of $PO_4^{3-}$ in solution. The content of $PO_4^{3-}$ exhibited the same trend as $Fe^{3+}$, which further confirms the simultaneous release of $PO_4^{3-}$ and $Fe^{3+}$ by the action of *C. vulgaris*. In the iron-free group of the first cycle, the slight fluctuation of $PO_4^{3-}$ is primarily attributed to the combination of $Ca^{2+}$ and $Mg^{2+}$ in the medium. Other studies have shown that flocculating Fe–P can adsorb a part of $PO_4^{3-}$, which also increases the P content in the solution after the Fe–P is dissolved [18]. In the second cycle, the concentration of $PO_4^{3-}$ continued to decrease in all groups, which was ascribed to the adsorption effect of *C. vulgaris*. The difference in $PO_4^{3-}$ content remained stable after the 14th day because the P content of *C. vulgaris* cells was saturated under the high concentration of $PO_4^{3-}$.

To obtain the release amount of Fe–P ($\Delta Q$) under the influence of a certain concentration of microalgae, the most representative period in the first cycle was selected (3–5 d). Table 3 shows the

**Table 3.** The amount of Fe–P released in 2 d. ΔQ is the amount of Fe–P released in mg 200 ml$^{-1}$. The concentration of C. vulgaris is $5.6 \times 10^4$–$8 \times 10^4$ cell ml$^{-1}$.

| culture time | groups | | | | | |
|---|---|---|---|---|---|---|
| | 0.02 | 0.2 | 0.6 | 1 | 2 | 5 |
| 3 d | 0 | 0 | 0.32 | 0.73 | 1.36 | 1.37 |
| 5 d | 0 | 0 | 0.48 | 0.84 | 1.89 | 2.27 |
| ΔQ | 0 | 0 | 0.16 | 0.11 | 0.53 | 0.9 |
| | | | $0.11 \leq \Delta Q \leq 0.9$ | | | |

amount of Fe–P released on the third and fifth days; the range of Fe–P released in 2 d was 0.11–0.9 mg, and the amount of daily release was 0.055–0.45 mg.

## 3.2. Effect of Fe$^{3+}$ release on biomass, specific growth rate and pH

The amount of released Fe$^{3+}$ was the only factor that affected C. vulgaris growth, specific growth rate and pH value based on the same initial pH, light and other conditions in the second stage. Additionally, $PO_4^{3-}$ and Fe$^{3+}$ exhibited similar effects on the growth of C. vulgaris. Thus, either trace or excess $PO_4^{3-}$ and Fe$^{3+}$ inhibited the proliferation of C. vulgaris [32]. Figure 3 shows the changes in the biomass of C. vulgaris over three cycles. The growth cycle of C. vulgaris was 7 d (one logarithmic period), and the experiment was carried out for one month (four logarithmic periods). According to the results (figure 3), the three cycles of the interaction, Fe$^{3+}$ release and outbreak are divided, as shown in figures 1 and 2.

In the first cycle (figure 3a), the biomass of C. vulgaris remained constant with increasing amounts of Fe–P. In particular, the lowest biomass values were observed in the groups with more Fe–P (2 and 5 mg). With the sustained growth of C. vulgaris, the inhibitory effect of the iron-free group in the second cycle (figure 3b) was robust. Compared with other groups in the third cycle (figure 3c), the inhibitory effects of iron-free and 5 mg Fe–P groups on C. vulgaris became more significant. Therefore, 0.02–2 mg Fe–P was more beneficial to the growth of C. vulgaris in 200 ml water.

Similarly, the specific growth rate of C. vulgaris showed a significant difference under the long-term effects of different amounts of Fe–P. Table 4 summarizes the changes in the specific growth rate of C. vulgaris. In the first cycle, the specific growth rate of the 5 mg group reached 0.15 d$^{-1}$, whereas that of the other groups was greater than 0.17 d$^{-1}$. In the second cycle, the specific growth rate of 0 and 5 mg Fe–P was 0.33 and 0.34 d$^{-1}$, respectively, which are the lowest values. Like the second cycle, the specific growth rate of the non-ferric C. vulgaris and 5 mg Fe–P groups exhibited the minimum values (0.5 and 0.48 d$^{-1}$, respectively) in the third cycle, whereas the others were greater than 0.52 d$^{-1}$. In sum, both the non-ferric and maximum Fe–P (5 mg) groups exhibited long-term inhibitory effects on the specific growth rate of C. vulgaris in the case of saturated P content. This is likely to cause potential harm to P removal under the negative influence of Fe–P. By contrast, the biodegradation of Fe–P can convert inorganic P into bioavailable P.

According to table 4, the optimal group analysis of the specific growth rate was conducted under the effect of Fe–P. The 0.02 mg Fe–P group reached 0.2 d$^{-1}$ during the release phase of Fe$^{3+}$, which was the highest value in all groups; the biomass of the group was also the highest, as shown in figure 3. In the second cycle, the specific growth rate of 0.6 and 1 mg Fe–P groups reached a maximum of 0.4 d$^{-1}$. Therefore, 0.6–1 mg Fe–P is the optimum value for biomass production at the adsorption stage. It is worth noting that the change in biomass is not obvious, although the amount of Fe–P (from 0.2 to 1 mg) increased fivefold because the influencing factors in the solution are sufficient; the inhibition is reflected in the iron-free and 5 mg groups. In the third cycle, the self-growth phase of C. vulgaris reached its maximum specific growth rate. As a result, the specific growth rate of C. vulgaris in the sediment release stage was 0.3–0.4 d$^{-1}$, and the highest level of 0.53 d$^{-1}$ was achieved for 0.02–2 mg Fe–P in the 200 ml solution.

The metabolites of C. vulgaris during the growth process can promote an increase of pH [33]. The change of pH over the three cycles is shown in figure 4. This result seems to indicate that the 5 mg group exhibited the strongest inhibitory effect on pH increase during the whole process. Especially in

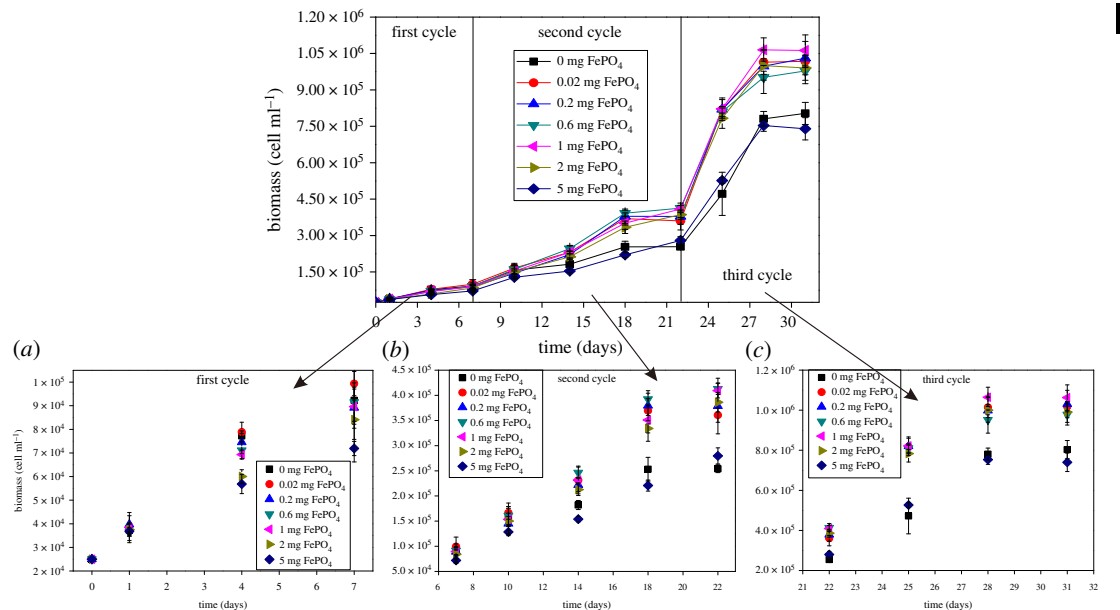

**Figure 3.** Changes of the biomass in three cycles with (*a*), (*b*) and (*c*) representing the first cycle, second cycle and third cycle, respectively. The difference in biomass production can be observed in 30 days of continuous cultivation. Data represent the mean and range of two independent experiments.

**Table 4.** Specific growth rate of microalgae in three cycles ($d^{-1}$). Standard error (s.e.) is displayed, $0.005 < s.e. < 0.1$.

| | specific growth rate ($d^{-1}$) | | | | | | |
|---|---|---|---|---|---|---|---|
| Fe–P (mg) | 0 | 0.02 | 0.2 | 0.6 | 1 | 2 | 5 |
| first cycle | $0.19 \pm 0.01$ | $0.20 \pm 0.01$ | $0.18 \pm 0.011$ | $0.19 \pm 0.007$ | $0.18 \pm 0.005$ | $0.1 \pm 0.014$ | $0.15 \pm 0.017$ |
| second cycle | $0.33 \pm 0.012$ | $0.38 \pm 0.013$ | $0.39 \pm 0.04$ | $0.4 \pm 0.034$ | $0.40 \pm 0.031$ | $0.3 \pm 0.012$ | $0.34 \pm 0.02$ |
| third cycle | $0.50 \pm 0.04$ | $0.53 \pm 0.02$ | $0.53 \pm 0.01$ | $0.52 \pm 0.14$ | $0.53 \pm 0.15$ | $0.53 \pm 0.1$ | $0.48 \pm 0.01$ |

the release phase of $Fe^{3+}$, the hydrolysis of $Fe^{3+}$ inhibits the pH change of the solution ($pH < 8$), whereas the iron-free group is greater than 8.8. Both the iron-free group and the 5 mg group showed lower pH levels in the second (pH of 9.97 and 9.61, respectively) and third cycle (pH of 10.87 and 10.46, respectively). The former is due to iron deficiency inhibition, while the latter is attributed to excessive inhibition. From the point of pH increase, the pH of the 5 mg Fe–P group showed the most balanced status, but this was not favourable for the growth of *C. vulgaris*, as shown in figure 3. By contrast, the promotion effect of the micro-groups (0.02–2 mg sediment) on algal growth is obvious despite the pH level. In addition, high pH values impaired the biomass properties and led to self-flocculation of the biomass, which is reflected in table 5.

## 3.3. Effect of release of $Fe^{3+}$ on flocculation efficiency

Flocculation efficiency is one of the factors affecting the concentration of suspended cells; it is mainly affected by pH and metal cations [34]. The agglomeration of the microalgae cells becomes stronger with higher efficiency, which is not conducive to the practical application of P removal.

Previous studies have shown that self-flocculation occurs when pH is higher than 10.5 [35]. Here, investigation of the flocculation effect is important for explaining the influence of $Fe^{3+}$ because flocculation is initiated by the interaction of $Fe^{3+}$ and *C. vulgaris*, especially in the second cycle. This requires a separate study of each stage; we combined the data from table 4 and figure 4. The pH of all groups was less than 10.5 from the first cycle to the early stage of the second cycle based on the results in figure 4. Therefore, flocculation is mainly affected by $Fe^{3+}$ released from Fe–P.

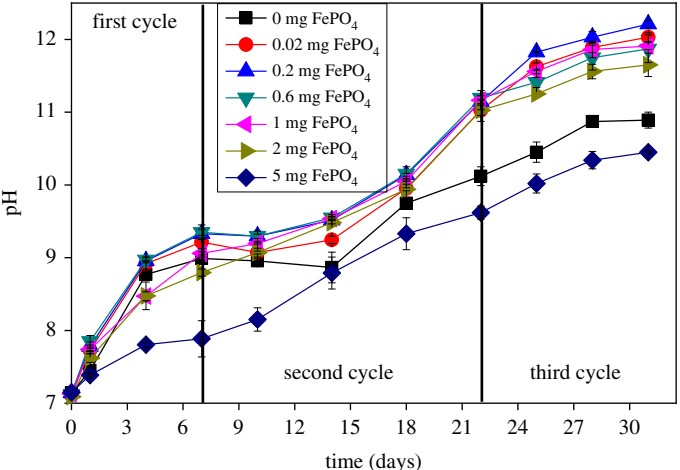

**Figure 4.** Changes of pH with different amounts of FePO$_4$ in the proliferation of C. vulgaris. Data represent the mean and range of two independent experiments.

**Table 5.** Flocculation efficiency (%) and rate of increase (%) during each cycle. Standard error (s.e.) is displayed, $0.7 <$ s.e. $< 6.4$.

| | Fe–P (mg) | 0 | 0.02 | 0.2 | 0.6 | 1 | 2 | 5 |
|---|---|---|---|---|---|---|---|---|
| first cycle | flocculation | 59.63 ± | 61.36 ± | 55.43 ± | 60.19 ± | 57.12 ± | 55.41 ± | 48.82 ± |
| | efficiency | 4.4 | 0.7 | 2.1 | 1.3 | 0.7 | 0.7 | 2.2 |
| | increment | 26.73 | 26.48 | 18.33 | 28.78 | 22.18 | 22.11 | 16.76 |
| second | flocculation | 63.67 ± | 72.43 ± | 76.46 ± | 77.83 ± | 78.12 ± | 78.23 ± | 74.29 ± |
| cycle | efficiency | 3.1 | 5.5 | 0.6 | 2.6 | 0.5 | 1.2 | 1.3 |
| | increment | 4.04 | 11.07 | 21.03 | 17.64 | 21 | 22.82 | 25.47 |
| third | flocculation | 68.37 ± | 64.58 ± | 63.27 ± | 57.79 ± | 60.36 ± | 61.00 ± | 62.21 ± |
| cycle | efficiency | 4.4 | 4.9 | 2.1 | 6.4 | 1.7 | 4.8 | 1.9 |
| | increment | 4.7 | −7.86 | −13.19 | −20.04 | −17.76 | −17.24 | −12.08 |

From the above results, it can be concluded that the flocculation efficiency of Fe$^{3+}$ released from Fe–P can be as high as 72–78%; however, this process only occurs during the release phase of Fe–P. In the short term, the growth of C. vulgaris is inhibited when there is a large amount of Fe–P, while P-pollution is enhanced. In the long term, the biodegradation of C. vulgaris on Fe–P has a cyclical pattern in the water-sediment system.

The flocculation efficiency and rate of increase during each cycle are listed in table 5. The flocculation efficiency of the iron-free medium reached a maximum of 68.37% during the third cycle because of the high pH ($10 <$ pH $< 10.9$). However, the highest flocculation efficiency of the Fe-containing medium in the second stage ranged from 72% to 78%, and the increment increased with the amount of Fe–P. This phase is consistent with the trend of the Fe$^{3+}$ release. In addition, the negative value for the increment indicates a drop in flocculation efficiency during the third cycle, which is closely related to the outbreak and high pH. Flocculation during the third cycle is mainly affected by self-flocculation without the influence of Fe$^{3+}$.

## 3.4. The change in removal efficiency of P

To investigate the effect of Fe–P release on P removal in solution, we determined the removal efficiency of P during each cycle, as shown in figure 5. The removal rate of P in the first cycle shows a rising negative value with an increase in the sediment. However, the removal efficiency is significantly improved when the positive value from the second cycle is larger than that of the third cycle (on average). This means that

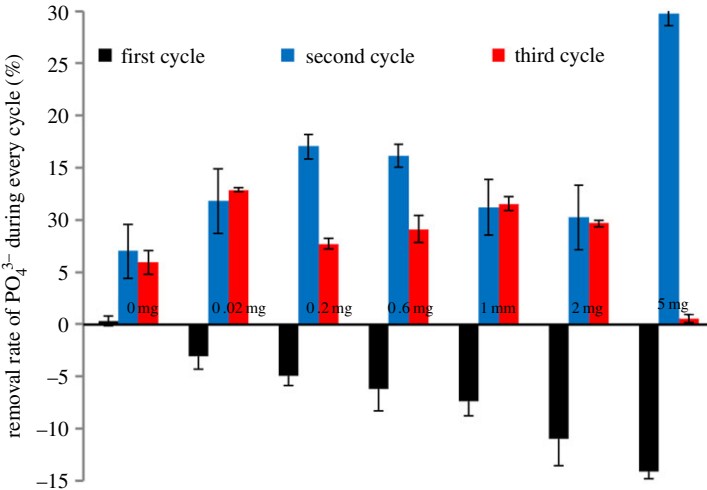

**Figure 5.** The removal rate of P during each cycle. The data are based on the initial and last time of each cycle. Data represent the mean and range of two independent experiments.

the presence of Fe–P and *C. vulgaris* can cause a temporary increase in P content. In the stage view, the decrease in the P removal rate could be caused by the internal P loading, and the removal rate was less than 30% during the co-cultivation of *C. vulgaris* and Fe–P. In the long term, phosphate biotransformation plays a key role in the ecological balance of the water–sediment system.

## 4. Conclusion

In this study, the effects of released $Fe^{3+}$ from Fe–P on biomass, flocculation and removal of $PO_4^{3-}$ were investigated under the action of *C. vulgaris*. The results indicated that *C. vulgaris* could promote the release of Fe–P, and the specific growth rate is lower than $0.2\,d^{-1}$ during the release period. The amount of Fe–P released was calculated, and the released amount ($\Delta Q$) reached $0.055$–$0.45\,mg\,d^{-1}$ in the 200 ml solution. $Fe^{3+}$ and $PO_4^{3-}$ are manifested in two ways. First, $Fe^{3+}$ from Fe–P can enhance the flocculation strength of *C. vulgaris* and inhibit biomass production; the flocculation efficiency of released $Fe^{3+}$ can be as high as 72–78%. Second, the P removal rate decreased significantly owing to internal P loading. This indicates that Fe–P is converted to bioavailable phosphorus by *C. vulgaris*. More research is warranted on this critical topic for the management of aquatic water bodies.

Data accessibility. All materials and data are available for download on the Dryad Digital Repository: https://doi.org/10.5061/dryad.5dv41ns6h [36].

Authors' contributions. L.H.: writing—original draft; Y.C.: conceptualization; S.C.: supervision; X.W.: writing—review and editing; J.L.: formal analysis.

All authors gave final approval for publication and agreed to be held accountable for the work performed therein.

Competing interests. The authors declare no conflict of interest.

Funding. This research was funded by the National Key Research and Development Plan of China (grant no. 2016TFC0502204) and the National Natural Science Foundation of China (grant no. 51809219).

Acknowledgements. We would like to thank the Southwest University of Science and Technology for providing the experimental facilities.

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
