## [Peer Review File · Royal Society Open Science]

Review History

RSOS-202198.R0 (Original submission)

Review form: Reviewer 1

Is the manuscript scientifically sound in its present form?

Yes

Are the interpretations and conclusions justified by the results?

Yes

Is the language acceptable?

Yes

Do you have any ethical concerns with this paper?

No

Have you any concerns about statistical analyses in this paper?

No

Recommendation?

Major revision is needed (please make suggestions in comments)

Comments to the Author(s)

Manuscript entitled "Effects of ferric phosphate deposit on biomass and phosphorus removal by consecutive cultivation with *Chlorella vulgaris*" dealt with the effects of Fe-P re-release on biomass, flocculation and removal of PO₄³⁻ by *Chlorella vulgaris*. The work is interesting and I would like to recommend its publication after some revisions.

Page 4 line 41. "Therefore, *C. vulgaris* have been used in the treatment of P-contaminated water bodies, such as *Chlorella vulgaris* (*C. vulgaris*)" Please correct this sentence.

Page 4 line 53. "PO₄³⁻ is often combined with Fe³⁺ to form solid-phase compound (Ferric phosphate, Fe-P) that precipitate or adsorb to river sediments with the pollution expands on the timescale. The sentence is not clear, please correct.

Please increase Figure 3 resolution. Characters are difficult to read.

All figures. The lines that authors use to distinct the different cycles cover the symbols and the error bars. You may correct this by using, for instance, a thinner line.

A more detailed discussion should be added. Please add a performance comparison with other similar works for displaying the advantages of this work.

Review form: Reviewer 2**Is the manuscript scientifically sound in its present form?**

Yes

Are the interpretations and conclusions justified by the results?

Yes

Is the language acceptable?

No

Do you have any ethical concerns with this paper?

No

Have you any concerns about statistical analyses in this paper?

No

Recommendation?

Major revision is needed (please make suggestions in comments)

Comments to the Author(s)

I cannot well understand what the authors told to us, a terrible writing style. The authors need to reorganize the manuscript by a fluent manner. Did you want to study the precipitation of Fe-P or the effects of *C. vulgaris* on P release from Fe-P in sediment? P release to solution is not a desirable result, so what is the meaning of P removal by consecutive cultivation with *C. vulgaris*? I don't think the authors have told us a clear story. The authors are recommended to rewrite the manuscript, including the title the manuscript, which is also not clear. For example, "effects of ^^^ on biomass", which biomass? Pls check the grammar issues of the manuscript carefully.

1. The English grammar in this paper must be improved for the reconsideration of publication. The manuscript needs to be edited for grammar and syntax.

2. The introduction was not well organized, pls reorganize this section by a logical way. At present version, we don't understand the main novelty of the work.
3. Line 17. The full name of *C. vulgaris* should be given.
4. Line 38. phosphorous? Pls check the similar mistakes throughout the manuscript.
5. Line 42. *Chlorella vulgaris* (*C. vulgaris*)?
6. Line 50. phosphate should be phosphate (PO₄³⁻).
7. Line 59. biology?
8. Line 66. but three general theories?
9. Line 77. Re?
10. Line 120. As shown in Table 1, all the spiked Fe³⁺ was transformed to Fe-P?
11. Line 204. this stage can be regarded as the adsorption of *C. vulgaris*?
12. Line 230. "The amount of released Fe³⁺ is the only factor that affects the *C. vulgaris* growth", the algae only need Fe³⁺ to sustain their growth?
13. Line 323. "To investigate the effect of Fe-P release on P removal", removed from the solution or sediment? It is hard to understand the authors' opinions.

Decision letter (RSOS-202198.R0)

Dear Mr Chen:

Title: Effects of ferric phosphate deposit on biomass and phosphorus removal by consecutive cultivation with *Chlorella vulgaris*
Manuscript ID: RSOS-202198

Thank you for your submission to Royal Society Open Science. The chemistry content of Royal Society Open Science is published in collaboration with the Royal Society of Chemistry. I apologise for the delay in sending you this decision.

The editor assigned to your manuscript has now received comments from reviewers. We would like you to revise your paper in accordance with the referee and Subject Editor suggestions which can be found below (not including confidential reports to the Editor). Please note this decision does not guarantee eventual acceptance.

Please submit your revised paper before 03-Jul-2021. Please note that the revision deadline will expire at 00.00am on this date. If we do not hear from you within this time then it will be assumed that the paper has been withdrawn. In exceptional circumstances, extensions may be possible if agreed with the Editorial Office in advance. We do not allow multiple rounds of revision so we urge you to make every effort to fully address all of the comments at this stage. If deemed necessary by the Editors, your manuscript will be sent back to one or more of the original reviewers for assessment. If the original reviewers are not available we may invite new reviewers.

On behalf of the Subject Editor Professor Anthony Stace and the Associate Editor Dr Nadia Martinez Villegas.

RSC Associate Editor:
Comments to the Author:

The research presented in this draft is original and of interest to RSOS audience, however a professional proofreading service is highly recommended as the language needs a thorough revision in order to communicate your research clearly. For example, no sentences are provided in Section 2.2, which makes it difficult to understand it. Additionally, the quality of the figures must be improved, especially to avoid repetition. Please read carefully each of the comments from the reviewers and address each of them.

RSC Subject Editor:
Comments to the Author:
(There are no comments.)

Reviewers' Comments to Author:

Reviewer: 1

Comments to the Author(s)

Manuscript entitled "Effects of ferric phosphate deposit on biomass and phosphorus removal by consecutive cultivation with *Chlorella vulgaris*" dealt with the effects of Fe-P re-release on biomass, flocculation and removal of PO₄³⁻ by *Chlorella vulgaris*. The work is interesting and I would like to recommend its publication after some revisions.

Page 4 line 41. "Therefore, *C. vulgaris* have been used in the treatment of P-contaminated water bodies, such as *Chlorella vulgaris* (*C. vulgaris*)" Please correct this sentence.

Page 4 line 53. "PO₄³⁻ is often combined with Fe³⁺ to form solid-phase compound (Ferric phosphate, Fe-P) that precipitate or adsorb to river sediments with the pollution expands on the timescale. The sentence is not clear, please correct.

Please increase Figure 3 resolution. Characters are difficult to read.

All figures. The lines that authors use to distinct the different cycles cover the symbols and the error bars. You may correct this by using, for instance, a thinner line.
A more detailed discussion should be added. Please add a performance comparison with other similar works for displaying the advantages of this work.

Reviewer: 2

Comments to the Author(s)

I cannot well understand what the authors told to us, a terrible writing style. The authors need to reorganize the manuscript by a fluent manner. Did you want to study the precipitation of Fe-P or the effects of *C. vulgaris* on P release from Fe-P in sediment? P release to solution is not a desirable result, so what is the meaning of P removal by consecutive cultivation with *C. vulgaris*? I don't think the authors have told us a clear story. The authors are recommended to rewrite the manuscript, including the title the manuscript, which is also not clear. For example, "effects of ^^^ on biomass", which biomass? Pls check the grammar issues of the manuscript carefully.

1. The English grammar in this paper must be improved for the reconsideration of publication. The manuscript needs to be edited for grammar and syntax.
2. The introduction was not well organized, pls reorganize this section by a logical way. At present version, we don't understand the main novelty of the work.
3. Line 17. The full name of *C. vulgaris* should be given.
4. Line 38. phosphorous? Pls check the similar mistakes throughout the manuscript.
5. Line 42. *Chlorella vulgaris* (*C. vulgaris*)?
6. Line 50. phospho should be phosphate (PO₄³⁻).
7. Line 59. biology?
8. Line 66. but three general theories?
9. Line 77. Re?
10. Line 120. As shown in Table 1, all the spiked Fe³⁺ was transformed to Fe-P?
11. Line 204. this stage can be regarded as the adsorption of *C. vulgaris*?
12. Line 230. "The amount of released Fe³⁺ is the only factor that affects the *C. vulgaris* growth", the algae only need Fe³⁺ to sustain their growth?
13. Line 323. "To investigate the effect of Fe-P release on P removal", removed from the solution or sediment? It is hard to understand the authors' opinions.

Author's Response to Decision Letter for (RSOS-202198.R0)

See Appendices A & B.

Decision letter (RSOS-202198.R1)

Dear Mr Chen:

Title: Effects of *Chlorella vulgaris* on P release from ferric phosphate sediment by consecutive cultivation

Manuscript ID: RSOS-202198.R1

Thank you for your submission, your above manuscript has now been reviewed. The comments from reviewers are included at the bottom of this letter.

In view of the criticisms of the reviewers, the manuscript has been rejected in its current form. However, a new manuscript may be submitted which takes into consideration these comments.

Please note that resubmitting your manuscript does not guarantee eventual acceptance, and that your resubmission will be subject to peer review before a decision is made.

Your resubmitted manuscript should be submitted by 14-Feb-2022. If you are unable to submit by this date please contact the Editorial Office.

Sincerely,
Dr Laura Smith
Publishing Editor, Journals

On behalf of the Subject Editor Professor Anthony Stace and the Associate Editor Dr Nadia Martinez Villegas.

RSC Associate Editor

Comments to the Author:

Authors of this manuscript failed to achieve clear, fluent and correct English grammar and syntax. Several English errors persist in the draft and were also found in the responses to reviewers.

For example, the authors use the word 'sediment' to refer to iron phosphate. However, strictly speaking, iron phosphate is not a sediment, a sediment is a much more complex system than a single compound. The authors also use the word 'structure' instead of 'species' to refer to PO₄³⁻, which is incorrect. They use the verb 'to reduce' instead of 'to decrease' for a species that reduces (Fe³⁺), which makes the reading confusing, and so on.

Author's Response to Decision Letter for (RSOS-202198.R1)

See Appendix C.

RSOS-211391.R0

Review form: Reviewer 1

Is the manuscript scientifically sound in its present form?

Yes

Are the interpretations and conclusions justified by the results?

Yes

Is the language acceptable?

Yes

Do you have any ethical concerns with this paper?

No

Have you any concerns about statistical analyses in this paper?

No

Recommendation?

Accept with minor revision (please list in comments)

Comments to the Author(s)

Manuscript entitled "Effects of *Chlorella vulgaris* on P release from ferric phosphate sediment by consecutive cultivation" dealt with the effects of Fe-P re-release on biomass, flocculation and removal of PO₄³⁻ by *Chlorella vulgaris*. The work is interesting and I would like to recommend its publication after some revisions.

The word "sediment" is missing from the title in the manuscript.

Please increase figures resolution. For instance, in figure 3 characters are difficult to read.

Review form: Reviewer 2

Is the manuscript scientifically sound in its present form?

Yes

Are the interpretations and conclusions justified by the results?

Yes

Is the language acceptable?

Yes

Do you have any ethical concerns with this paper?

No

Have you any concerns about statistical analyses in this paper?

No

Recommendation?

Major revision is needed (please make suggestions in comments)

Comments to the Author(s)

The authors need to provide a point-to-point response file to us.

Decision letter (RSOS-211391.R0)

Dear Mr Chen:

Title: Effects of *Chlorella vulgaris* on P release from ferric phosphate sediment by consecutive cultivation

Manuscript ID: RSOS-211391

The editor assigned to your paper has now received comments from reviewers. We would like you to revise your paper in accordance with the referee and Subject Editor suggestions which can be found below (not including confidential reports to the Editor). Please note this decision does not guarantee eventual acceptance.

Please submit a copy of your revised paper before 30-Oct-2021. Please note that the revision deadline will expire at 00.00am on this date. If we do not hear from you within this time then it will be assumed that the paper has been withdrawn. In exceptional circumstances, extensions may be possible if agreed with the Editorial Office in advance. We do not allow multiple rounds of revision so we urge you to make every effort to fully address all of the comments at this stage. If deemed necessary by the Editors, your manuscript will be sent back to one or more of the original reviewers for assessment. If the original reviewers are not available we may invite new reviewers.

Yours sincerely,
Dr Ellis Wilde

Publishing Editor, Journals

On behalf of the Subject Editor Professor Anthony Stace and the Associate Editor Dr Nadia Martinez Villegas.

RSC Associate Editor
 Comments to the Author:
 Dear authors,

We would like you to revise your paper (point by point) in accordance with the referee and Subject Editor suggestions made to your previous manuscript (RSOS-202198), which can be found below (not including confidential reports to the Editor).

RSC Associate Editor:
 Comments to the Author:

The research presented in this draft is original and of interest to RSOS audience, however a professional proofreading service is highly recommended as the language needs a thorough revision in order to communicate your research clearly. For example, no sentences are provided in Section 2.2, which makes it difficult to understand it. Additionally, the quality of the figures must be improved, especially to avoid repetition. Please read carefully each of the comments from the reviewers and address each of them.

RSC Subject Editor:
 Comments to the Author:
 (There are no comments.)

Reviewers' Comments to Author:
 Reviewer: 1

Comments to the Author(s)

Manuscript entitled "Effects of ferric phosphate deposit on biomass and phosphorus removal by consecutive cultivation with *Chlorella vulgaris*" dealt with the effects of Fe-P re-release on biomass, flocculation and removal of PO₄³⁻ by *Chlorella vulgaris*. The work is interesting and I would like to recommend its publication after some revisions.

Page 4 line 41. "Therefore, *C. vulgaris* have been used in the treatment of P-contaminated water bodies, such as *Chlorella vulgaris* (*C. vulgaris*)" Please correct this sentence.

Page 4 line 53. "PO₄³⁻ is often combined with Fe³⁺ to form solid-phase compound (Ferric phosphate, Fe-P) that precipitate or adsorb to river sediments with the pollution expands on the timescale. The sentence is not clear, please correct.

Please increase Figure 3 resolution. Characters are difficult to read.

All figures. The lines that authors use to distinct the different cycles cover the symbols and the error bars. You may correct this by using, for instance, a thinner line.

A more detailed discussion should be added. Please add a performance comparison with other similar works for displaying the advantages of this work.

Reviewer: 2

Comments to the Author(s)

I cannot well understand what the authors told to us, a terrible writing style. The authors need to reorganize the manuscript by a fluent manner. Did you want to study the precipitation of Fe-P or the effects of *C. vulgaris* on P release from Fe-P in sediment? P release to solution is not a desirable result, so what is the meaning of P removal by consecutive cultivation with *C. vulgaris*? I don't think the authors have told us a clear story. The authors are recommended to rewrite the manuscript, including the title the manuscript, which is also not clear. For example, "effects of ^^^ on biomass", which biomass? Pls check the grammar issues of the manuscript carefully.

1. The English grammar in this paper must be improved for the reconsideration of publication. The manuscript needs to be edited for grammar and syntax.
2. The introduction was not well organized, pls reorganize this section by a logical way. At present version, we don't understand the main novelty of the work.
3. Line 17. The full name of *C. vulgaris* should be given.
4. Line 38. phosphorous? Pls check the similar mistakes throughout the manuscript.
5. Line 42. *Chlorella vulgaris* (*C. vulgaris*)?
6. Line 50. phosphate should be phosphate (PO₄³⁻).
7. Line 59. biology?
8. Line 66. but three general theories?
9. Line 77. Re?
10. Line 120. As shown in Table 1, all the spiked Fe³⁺ was transformed to Fe-P?
11. Line 204. this stage can be regarded as the adsorption of *C. vulgaris*?
12. Line 230. "The amount of released Fe³⁺ is the only factor that affects the *C. vulgaris* growth", the algae only need Fe³⁺ to sustain their growth?
13. Line 323. "To investigate the effect of Fe-P release on P removal", removed from the solution or sediment? It is hard to understand the authors' opinions.

Reviewers' Comments to Author:

Reviewer: 2

Comments to the Author(s)

The authors need to provide a point-to-point response file to us.

Reviewer: 1

Comments to the Author(s)

Manuscript entitled "Effects of *Chlorella vulgaris* on P release from ferric phosphate sediment by consecutive cultivation" dealt with the effects of Fe-P re-release on biomass, flocculation and removal of PO₄³⁻ by *Chlorella vulgaris*. The work is interesting and I would like to recommend its publication after some revisions.

The word "sediment" is missing from the title in the manuscript.

Please increase figures resolution. For instance, in figure 3 characters are difficult to read.

Author's Response to Decision Letter for (RSOS-211391.R0)

See Appendix D - F.

RSOS-211391.R1

Review form: Reviewer 1

Is the manuscript scientifically sound in its present form?

Yes

Are the interpretations and conclusions justified by the results?

Yes

Is the language acceptable?

Yes

Do you have any ethical concerns with this paper?

No

Have you any concerns about statistical analyses in this paper?

No

Recommendation?

Accept as is

Comments to the Author(s)

The authors performed the requested changes and corrections. The paper is now suitable for publication.

Review form: Reviewer 2

Is the manuscript scientifically sound in its present form?

Yes

Are the interpretations and conclusions justified by the results?

Yes

Is the language acceptable?

No

Do you have any ethical concerns with this paper?

No

Have you any concerns about statistical analyses in this paper?

No

Recommendation?

Major revision is needed (please make suggestions in comments)

Comments to the Author(s)

Two question,

- 1) The authors claim that they have polished the grammar, but I still find lots of sloppy English. For example, "The metabolites of *C. vulgaris* during the growth process can promote an increase in pH"? "in" should be "of". Pls improve the manuscript carefully again.
- 2) In the first round of review, I recommend the authors to correct the mistakes of the use of "full names" and (or) "abbreviated names", which don't mean that you need to give the full name of abbreviated name in the bracket following their abbreviated name or full name. For the 1st use of a Latin name of certain organism, you should use the full name, while for the 2nd use you can use the abbreviated name.

Decision letter (RSOS-211391.R1)

Dear Mr Chen:

Title: Effects of *Chlorella vulgaris* on P release from ferric phosphate sediment by consecutive cultivation

Manuscript ID: RSOS-211391.R1

The editor assigned to your paper has now received comments from reviewers. We would like you to revise your paper in accordance with the referee and Subject Editor suggestions which can be found below (not including confidential reports to the Editor). Please note this decision does not guarantee eventual acceptance.

Please submit a copy of your revised paper before 12-Jan-2022. Please note that the revision deadline will expire at 00.00am on this date. If we do not hear from you within this time then it will be assumed that the paper has been withdrawn. In exceptional circumstances, extensions may be possible if agreed with the Editorial Office in advance. We do not allow multiple rounds of revision so we urge you to make every effort to fully address all of the comments at this stage. If deemed necessary by the Editors, your manuscript will be sent back to one or more of the original reviewers for assessment. If the original reviewers are not available we may invite new reviewers.

Please also include the following statements alongside the other end statements. As we cannot publish your manuscript without these end statements included, if you feel that a given heading is not relevant to your paper, please nevertheless include the heading and explicitly state that it is not relevant to your work.

- Ethics statement

Please clarify whether you received ethical approval from a local ethics committee to carry out your study. If so please include details of this, including the name of the committee that gave consent in a Research Ethics section after your main text. Please also clarify whether you received informed consent for the participants to participate in the study and state this in your Research Ethics section.

OR

Please clarify whether you obtained the necessary licences and approvals from your institutional animal ethics committee before conducting your research. Please provide details of these licences and approvals in an Animal Ethics section after your main text.

OR

Please clarify whether you obtained the appropriate permissions and licences to conduct the fieldwork detailed in your study. Please provide details of these in your methods section.

- Data accessibility

It is a condition of publication that you make available the data and research materials supporting the results in the article. Datasets should be deposited in an appropriate publicly available repository and details of the associated accession number, link or DOI to the datasets must be included in the Data Accessibility section of the article (<https://royalsocietypublishing.org/rsos/for-authors#question17>). Reference(s) to datasets should also be included in the reference list of the article with DOIs (where available).

Please include a Data Availability section after your main text stating where supporting data are available from, or where they will be made available should your article be accepted for publication.

If you wish to submit your supporting data or code to Dryad (<http://datadryad.org/>), or modify your current submission to dryad, please use the following link:
<http://datadryad.org/submit?journalID=RSOS&manu=RSOS-211391.R1>

- Competing interests

Please include a Competing Interests section after your main text declaring any financial or non-financial competing interests. If you have no competing interests please state 'I/we have no competing interests.'

- Authors' contributions

Please include an Authors' Contributions section at the end of your main text detailing the contribution of each author. All authors should have read and approved the manuscript before submission and this should be stated in the Authors' Contributions section.

The list of Authors should meet all of the following criteria; 1) substantial contributions to conception and design, or acquisition of data, or analysis and interpretation of data; 2) drafting the article or revising it critically for important intellectual content; and 3) final approval of the version to be published.

- Acknowledgements

- Funding statement

Please include a funding section after your main text which lists the source of funding for each author.

Yours sincerely,
Dr Ellis Wilde
Publishing Editor, Journals

On behalf of the Subject Editor Professor Anthony Stace and the Associate Editor Dr Nadia Martinez Villegas.

RSC Associate Editor

Comments to the Author:

In view of the criticisms of a reviewer, found at the bottom of this letter, two major issues still need to be addressed. Please read carefully each of the comments from the reviewer carefully and address each of them.

As suggested previously, a professional proofreading service is highly recommended to improve the language.

RSC Subject Editor

Comments to the Author:

(There are no comments.)

Reviewers' Comments to Author:

Reviewer: 2

Comments to the Author(s)

Two question,

- 1) The authors claim that they have polished the grammar, but I still find lots of sloppy English. For example, "The metabolites of *C. vulgaris* during the growth process can promote an increase in pH"? "in" should be "of". Pls improve the manuscript carefully again.
- 2) In the first round of review, I recommend the authors to correct the mistakes of the use of "full names" and (or) "abbreviated names", which don't mean that you need to give the full name of abbreviated name in the bracket following their abbreviated name or full name. For the 1st use of a Latin name of certain organism, you should use the full name, while for the 2nd use you can use the abbreviated name.

Reviewer: 1

Comments to the Author(s)

The authors performed the requested changes and corrections. The paper is now suitable for publication.

Author's Response to Decision Letter for (RSOS-211391.R1)

See Appendix G.

Decision letter (RSOS-211391.R2)

Dear Mr Chen:

Title: Effects of *Chlorella vulgaris* on P release from ferric phosphate sediment by consecutive cultivation

Manuscript ID: RSOS-211391.R2

It is a pleasure to accept your manuscript in its current form for publication in Royal Society Open Science. The chemistry content of Royal Society Open Science is published in collaboration with the Royal Society of Chemistry.

Yours sincerely,
Dr Ellis Wilde
Publishing Editor, Journals

RSC Associate Editor
Comments to the Author:
(There are no comments.)

Reviewer(s)' Comments to Author:

Appendix A

Response to Reviewer 1 Comments

Thank you for your professional reviews. The manuscript has been revised according to your suggestions.

Point 1: Page 4 line 41. "Therefore, *C. vulgaris* have been used in the treatment of P-contaminated water bodies, such as *Chlorella vulgaris* (*C. vulgaris*)" Please correct this sentence.

Response 1: This sentence has been deleted and a new addition (*C. vulgaris* have been used in the treatment of eutrophic water bodies) has been made in line 38 in order to make the expression clearer.

Point 2: Page 4 line 53. " PO_4^{3-} is often combined with Fe^{3+} to form solid-phase compound (Ferric phosphate, Fe-P) that precipitate or adsorb to river sediments with the pollution expands on the timescale. The sentence is not clear, please correct.

Response 2: The sentence has been modified: Iron phosphate (Fe-P) deposits are often formed by the combination of PO_4^{3-} and Fe^{3+} . As a common sediment in rivers and lakes, however, Fe-P is considered to be an active compound but has very high release rate.

Point 3: Please increase Figure 3 resolution. Characters are difficult to read.

Response 3: Figure 3 resolution has been increased.

Point 4: All figures. The lines that authors use to distinct the different cycles cover the symbols and the error bars. You may correct this by using, for instance, a thinner line.

Response 4: This suggestion has been adopted in the manuscript. All pictures have been adjusted according to your suggestions.

Point 5: A more detailed discussion should be added. Please add a performance comparison with other similar works for displaying the advantages of this work.

Response 5: Several discussions and references have been added to the abstract. Large-scale revisions are made in the abstract section

Introduction: Domestic sewage, agricultural effluent and industrial wastewater containing

large amounts of phosphorus (P) that are discharged into river water significantly promote the eutrophication of water bodies, which not only damages the ecosystems of the river but also harms aquatic plants and animals [1,2]. *Chlorella vulgaris* (*C. vulgaris*) propagate by using inorganic nutrients, including nitrogen (N) and phosphorus (P), which means it have a high potential for removing inorganic nutrients from surrounding waters. Therefore, *C. vulgaris* is used in the treatment of eutrophic water bodies [3]. From the perspective of biomaterials, *C. vulgaris* shows a high metal binding capacity because of the presence of polysaccharides and various functional groups of proteins or lipids on the cell wall surface (such as amino, hydroxyl, carboxyl, and sulfate) that can serve as binding sites for metal ions [4]. Currently, *C. vulgaris* has been widely used in wastewater (domestic sewage, industrial wastewater, and municipal wastewater) treatment based on its optimal adsorption or absorption capacity for N, P and metal cations (such as Fe^{3+} , Cr^{5+} , Ni^{2+} , Zn^{2+} , Cu^{2+} , and Pb^{2+}) during its proliferation process [5-10].

P in river and lake is mainly present as PO_4^{3-} [11], this structure can be readily combined with the metal cations (such as Fe^{3+} , Ca^{2+} and Al^{3+}) to form a precipitate and stored in the sediment of the river [12]. Iron phosphate (Fe-P) deposits are often formed by the combination of PO_4^{3-} and Fe^{3+} . The precipitate has the most important effect on the overlying water because of persistent release and periodic outbreaks of microalgae [13]. As a common sediment in rivers and lakes, however, Fe-P is considered to be an active compound but has very high release rate [14]. The release of PO_4^{3-} and cations from the sediment could cause persistent pollution of upper water by the actions of microorganisms or human disturbance, which has become a potential hazard to the water environment [15,16]. Zhu et al. initially studied the internal nutrient release responds to algal bloom decay from Taihu Lake in China, the research shows sediments from the river mouth released P to the overlying water in the presence of blooms [17].

For now, the mechanism of Fe-P release under the action of *C. vulgaris* has not been clarified. There are several mechanisms that may be used to explain the release of Fe-P. The first is that the reductive dissolution is caused under the anoxic conditions of sediment-water interface by the respiration of *C. vulgaris*. Second, the competitive behavior of OH^- and PO_4^{3-} may be taken as the second part. Third, the reverse reaction of Fe^{3+} hydrolysis is enhanced by a weak acid environment [18,19].

Release of excess Fe^{3+} from Fe-P sediment can inhibit the growth of *C. vulgaris* and even exhibits toxicity effect, which is unfavorable for treatment of eutrophic wastewater [22,23]. Periodical algal bloom is induced by the release of P from the P-containing sediment [24,25]. In addition, Fe^{3+} has a strong flocculation effect on *C. vulgaris*, because the surface of *C. vulgaris* is negatively charged and evenly suspended in liquid [20,21]. Therefore, chain reaction involving biomass production, pH fluctuation, flocculation, co-precipitation, and removal of PO_4^{3-} can be triggered through the release dynamics of Fe-P sediment. However, these processes have not been employed to evaluate the advantages and disadvantages of *C. vulgaris*.

In practical applications, phosphorus contaminated water-body is repaired by *C. vulgaris* in an Fe-P rich environment. To simulate actual environmental conditions, in this study, using different amounts of FeCl_3 and high content K_2HPO_4 to quantify Fe-P flocculated sediment, the sediment was cultured in BG-11 medium (removed PO_4^{3-} and ferric ammonium) with *C. vulgaris* during a continuous three-cycle culture. Effects of $\text{Fe}^{3+}/\text{PO}_4^{3-}$ release from Fe-P on

biomass, PO₄³⁻ removal and flocculation were investigated. Finally, this paper evaluates the environmental effects of *C. vulgaris* in the Fe-P rich sediment rivers.

Appendix B

Response to Reviewer 2 Comments

Reviewer 2:

I cannot well understand what the authors told to us, a terrible writing style. The authors need to reorganize the manuscript by a fluent manner. Did you want to study the precipitation of Fe-P or the effects of *C. vulgaris* on P release from Fe-P in sediment? P release to solution is not a desirable result, so what is the meaning of P removal by consecutive cultivation with *C. vulgaris*? I don't think the authors have told us a clear story. The authors are recommended to rewrite the manuscript, including the title the manuscript, which is also not clear. For example, "effects of ^^^ on biomass", which biomass? Pls check the grammar issues of the manuscript carefully.

Author response:

Special thanks for your professional comments, this is very helpful for me, because the paper does have the problem you said. I have made changes based on your suggestions and comments as much as possible, and at the same time, made new adjustments to the unclear areas. Thank you for your work. I hope the revised version can be approved.

This paper has been reorganized, in order to make the article more logical. Rewritten for more concise and coherence. At the same time, a new title (Effects of *Chlorella vulgaris* on P release from ferric phosphate sediment by consecutive cultivation) is created. In addition, the grammar issues have been corrected.

The subject of the article: *Chlorella vulgaris* is used to remove phosphorus from aqueous solutions in Fe-P sediment environment. There is a question here, is the phosphorus in the solution really reduced? At the same time, how does the amount of Fe-P and biomass change? In fact, the interaction between Fe-P and *Chlorella vulgaris* is reflected in the process of co-cultivation. This paper provides an important reference for the application of *Chlorella vulgaris* in an Fe-P rich environment.

Point to Point

Point 1: The English grammar in this paper must be improved for the reconsideration of publication. The manuscript needs to be edited for grammar and syntax.

Response 1: For this article, I have invited professionals to revise the grammar. At the same time, some sentences were added and deleted, and the sentence structure was adjusted in order to make the whole article more logical.

Point 2: The introduction was not well organized, pls reorganize this section by a logical way. At present version, we don't understand the main novelty of the work.

Response 2: The introduction has been reorganized, in order to make the article more logical.

Introduction: Domestic sewage, agricultural effluent and industrial wastewater

containing large amounts of phosphorus (P) that are discharged into river water significantly promote the eutrophication of water bodies, which not only damages the ecosystems of the river but also harms aquatic plants and animals [1,2]. *Chlorella vulgaris* (*C. vulgaris*) propagate by using inorganic nutrients, including nitrogen (N) and phosphorus (P), which means it have a high potential for removing inorganic nutrients from surrounding waters. Therefore, *C. vulgaris* is used in the treatment of eutrophic water bodies [3]. From the perspective of biomaterials, *C. vulgaris* shows a high metal binding capacity because of the presence of polysaccharides and various functional groups of proteins or lipids on the cell wall surface (such as amino, hydroxyl, carboxyl, and sulfate) that can serve as binding sites for metal ions [4]. Currently, *C. vulgaris* has been widely used in wastewater (domestic sewage, industrial wastewater, and municipal wastewater) treatment based on its optimal adsorption or absorption capacity for N, P and metal cations (such as Fe^{3+} , Cr^{5+} , Ni^{2+} , Zn^{2+} , Cu^{2+} , and Pb^{2+}) during its proliferation process [5-10].

P in river and lake is mainly present as PO_4^{3-} [11], this structure can be readily combined with the metal cations (such as Fe^{3+} , Ca^{2+} and Al^{3+}) to form a precipitate and stored in the sediment of the river [12]. Iron phosphate (Fe-P) deposits are often formed by the combination of PO_4^{3-} and Fe^{3+} . The precipitate has the most important effect on the overlying water because of persistent release and periodic outbreaks of microalgae [13]. As a common sediment in rivers and lakes, however, Fe-P is considered to be an active compound but has very high release rate [14]. The release of PO_4^{3-} and cations from the sediment could cause persistent pollution of upper water by the actions of microorganisms or human disturbance, which has become a potential hazard to the water environment [15,16]. Zhu et al. initially studied the internal nutrient release responds to algal bloom decay from Taihu Lake in China, the research shows sediments from the river mouth released P to the overlying water in the presence of blooms [17].

For now, the mechanism of Fe-P release under the action of *C. vulgaris* has not been clarified. There are several mechanisms that may be used to explain the release of Fe-P. The first is that the reductive dissolution is caused under the anoxic conditions of sediment-water interface by the respiration of *C. vulgaris*. Second, the competitive behavior of OH^- and PO_4^{3-} may be taken as the second part. Third, the reverse reaction of Fe^{3+} hydrolysis is enhanced by a weak acid environment [18,19].

Release of excess Fe^{3+} from Fe-P sediment can inhibit the growth of *C. vulgaris* and even exhibits toxicity effect, which is unfavorable for treatment of eutrophic wastewater [22,23]. Periodical algal bloom is induced by the release of P from the P-containing sediment [24,25]. In addition, Fe^{3+} has a strong flocculation effect on *C. vulgaris*, because the surface of *C. vulgaris* is negatively charged and evenly suspended in liquid [20,21]. Therefore, chain reaction involving biomass production, pH fluctuation, flocculation, co-precipitation, and removal of PO_4^{3-} can be triggered through the release dynamics of Fe-P sediment. However, these processes have not been employed to evaluate the advantages and disadvantages of *C. vulgaris*.

In practical applications, phosphorus contaminated water-body is repaired by *C. vulgaris* in an Fe-P rich environment. To simulate actual environmental conditions, in this study, using different amounts of FeCl_3 and high content K_2HPO_4 to quantify Fe-P

flocculated sediment, the sediment was cultured in BG-11 medium (removed PO_4^{3-} and ferric ammonium) with *C. vulgaris* during a continuous three-cycle culture. Effects of $\text{Fe}^{3+}/\text{PO}_4^{3-}$ release from Fe-P on biomass, PO_4^{3-} removal and flocculation were investigated. Finally, this paper evaluates the environmental effects of *C. vulgaris* in the Fe-P rich sediment rivers.

Point 3: Line 17. The full name of *C. vulgaris* should be given.

Response 3: The full name of *C. vulgaris* (*Chlorella vulgaris*) has been modified in the abstract section.

Point4: Line 38. phosphorous? Pls check the similar mistakes throughout the manuscript.

Response 4: The word error has been corrected. It should be phosphorus. At the same time, other similar problems were checked and corrected.

Point 5: Line 42. *Chlorella vulgaris* (*C. vulgaris*)?

Response 5: *Chlorella vulgaris* is abbreviated as *C. vulgaris* or *C. v.* The former is used in this article. It is based on the results of many articles (Uniform writing of many references).

Point 6: Line 50. phosphate should be phosphate (PO_4^{3-}).

Response 6: “phosphate” is replaced by “ PO_4^{3-} ”.

Point 7: Line 59. biology?

Response 7: This word has been deleted.

Point 8: Line 66. but three general theories?

Response 8: This sentence has been replaced by a new one: There are several mechanisms that may be used to explain the release of Fe-P.

Point 9: Line 77. Re?

Response 9: This word (Re-release) has been modified. “Re-” has been deleted.

Point 10: Line 120. As shown in Table 1, all the spiked Fe^{3+} was transformed to Fe-P?

Response 10: The concentration of PO_4^{3-} in the medium is as high as 60mg/L. However, the group with the highest Fe^{3+} concentration is 25mg/L, so all the spiked Fe^{3+} was

transformed to Fe-P during the experiment. If there is a little error, the change of the iron ion concentration in the solution shown in Figure 1 still has a reference degree according to the actual measurement. In addition, the error of the parallel sample shown in Figure 1 is also very small.

Point 11: Line 204. this stage can be regarded as the adsorption of *C. vulgaris*?

Response 11: Because Fe^{3+} in the solution are significantly reduced with the increase of flocculation efficiency in this stage. Simultaneously, the growth of biomass is weakly inhibited according to Figure 3b. The adsorption stage of microalgae is proved by the data of biomass and flocculation efficiency.

Point 12: Line 230. “The amount of released Fe^{3+} is the only factor that affects the *C. vulgaris* growth”, the algae only need Fe^{3+} to sustain their growth?

Response 12: Here is the author did not express clearly, further explanation has been added. This sentence means that other background conditions (such as pH temperature, light, BG-11, etc.) are the same. The amount of released Fe^{3+} becomes the dominant factor affecting the experimental process.

Point 13: Line 323. “To investigate the effect of Fe-P release on P removal”, removed from the solution or sediment? It is hard to understand the authors’ opinions.

Response 13: Here is the author did not express clearly. This sentence has been rephrased: To investigate the effect of Fe-P sediment release on P removal in solution.

Appendix C

Dear RSC Associate Editor,

We feel great thanks for your professional review work on our article. As you are concerned, there are several problems that need to be addressed. Due to my limited English ability, I made some language mistakes in the article. But I would like to re-submit this manuscript, and hope it is acceptable for publication in the journal.

According to your nice suggestions, we have made extensive corrections to our previous draft. Especially the language improvement.

1. We feel sorry for our carelessness. In our resubmitted manuscript, the typo is revised.
2. We tried our best to improve the manuscript and made some changes to the manuscript. These changes will not influence the content and framework of the paper (use the word 'sediment' to refer to iron phosphate; use the word 'structure' instead of 'species' to refer to PO_4^{3-} ; use the verb 'to reduce' instead of 'to decrease' for a species that reduces (Fe^{3+}) and so on) .
3. We did not list the changes but marked in red in the revised paper.

Sincerely,

Chen

Appendix D

Response to Reviewer 1 Comments

Thank you for your professional reviews. The manuscript has been revised according to your suggestions. The main corrections in the paper and the responds to the reviewer's comments are as flowing:

Point 1: Manuscript entitled "Effects of *Chlorella vulgaris* on P release from ferric phosphate sediment by consecutive cultivation" dealt with the effects of Fe-P re-release on biomass, flocculation and removal of PO₄³⁻ by *Chlorella vulgaris*. The work is interesting and I would like to recommend its publication after some revisions. The word "sediment" is missing from the title in the manuscript Please increase figures resolution. For instance, in figure 3 characters are difficult to read.

Response 1: Thank you for your reminder, "sediment" has been added to the title of the manuscript. The new title is "Effects of *Chlorella vulgaris* on P release from ferric phosphate sediment by consecutive cultivation". In addition, Figure 3 has been updated, the numbers on the ordinate are clearer, the resolution of the picture has also been improved.

Appendix E

Response to Reviewer 2 Comments

We thank you very much for giving us an opportunity to revise our manuscript, those comments are all valuable and very helpful for revising and improving our paper, as well as the important guiding significance to our researches. We have studied comments carefully and have made correction which we hope meet with approval. Revised portion are marked in red in the paper. The main corrections in the paper and the responds to the reviewer's comments are as flowing:

Reviewer 2:

I cannot well understand what the authors told to us, a terrible writing style. The authors need to reorganize the manuscript by a fluent manner. Did you want to study the precipitation of Fe-P or the effects of *C. vulgaris* on P release from Fe-P in sediment? P release to solution is not a desirable result, so what is the meaning of P removal by consecutive cultivation with *C. vulgaris*? I don't think the authors have told us a clear story. The authors are recommended to rewrite the manuscript, including the title the manuscript, which is also not clear. For example, "effects of ^^^ on biomass", which biomass? Pls check the grammar issues of the manuscript carefully.

Author response:

Special thanks for your professional comments, this is very helpful for me, because the paper does have the problem you said. I have made changes based on your suggestions and comments as much as possible, and at the same time, made new adjustments to the unclear areas. Thank you for your work. I hope the revised version can be approved.

This paper has been reorganized, in order to make the article more logical. Rewritten for more concise and coherence. At the same time, a new title (Effects of *Chlorella vulgaris* on P release from ferric phosphate sediment by consecutive cultivation) is created. In addition, grammar problems are corrected through professional platforms "Paperpal Preflight".

The subject of the article: *Chlorella vulgaris* is used to remove phosphorus from aqueous solutions in Fe-P sediment environment. There is a question here, is the phosphorus in the solution really reduced? At the same time, how does the amount of Fe-P and biomass change? In fact, the interaction between Fe-P and *Chlorella vulgaris* is reflected in the process of co-cultivation. This paper provides an important reference for the application of *Chlorella vulgaris* in an Fe-P rich environment.

Point to Point

Point 1: The English grammar in this paper must be improved for the reconsideration of publication. The manuscript needs to be edited for grammar and syntax.

Response 1: For this article, I have invited professionals to revise the grammar. Grammar issues have also been tested by professional platforms “Paperpal Preflight”. At the same time, some sentences were added and deleted, and the sentence structure was adjusted in order to make the whole article more logical. A large number of edits have been marked in red in the article.

Point 2: The introduction was not well organized, pls reorganize this section by a logical way. At present version, we don't understand the main novelty of the work.

Response 2: The introduction has been reorganized, in order to make the article more logical.

Introduction: Domestic sewage, agricultural effluent, and industrial wastewater containing large amounts of phosphorus (P) that are discharged into river water significantly promote the eutrophication of water bodies, which not only damages the ecosystems of the river but also harms aquatic plants and animals [1,2]. *Chlorella vulgaris* (*C. vulgaris*) propagates by using inorganic nutrients, including nitrogen (N) and phosphorus (P), which means it has a high potential for removing inorganic nutrients from the surrounding water. Therefore, *C. vulgaris* has been used for the treatment of eutrophic water bodies [3]. From the perspective of biomaterials, *C. vulgaris* shows a high metal binding capacity because of the presence of polysaccharides and various functional groups of proteins or lipids on the cell wall surface (such as amino, hydroxyl, carboxyl, and sulfate) that can serve as binding sites for metal ions [4]. Currently, *C. vulgaris* has been widely used in wastewater treatment (domestic sewage, industrial wastewater, and municipal wastewater) based on its optimal adsorption or absorption capacity for N, P, and metal cations (such as Fe^{3+} , Cr^{5+} , Ni^{2+} , Zn^{2+} , Cu^{2+} , and Pb^{2+}) during its proliferation process [5-10].

P in rivers and lakes is mainly present as PO_4^{3-} [11], which can be readily combined with metal cations (such as Fe^{3+} , Ca^{2+} , and Al^{3+}) to form a precipitate and stored in the sediment of the river [12]. Iron phosphate (Fe-P) can be constituted by a combination of PO_4^{3-} and Fe^{3+} in rivers and lakes. However, Fe-P is considered an active compound, but has a very high release rate [13]. It has the most important effect on the overlying water because of the persistent release and periodic outbreaks of microalgae [14]. The release of PO_4^{3-} and Fe^{3+} from Fe-P could cause persistent pollution of upper water by the actions of microorganisms or human disturbance, which has grown up to be a potential hazard to the water environment [15,16]. Zhu et al. initially studied the internal nutrient release response to algal bloom decay in Taihu Lake in China. Research has shown that sediment from the river mouth releases P to the overlying water in the presence of blooms [17].

To date, the mechanism of Fe-P release under the action of *C. vulgaris* has not been clarified. Several mechanisms may explain the release of Fe-P. First, reductive dissolution is caused by the respiration of *C. vulgaris* under anoxic conditions at the sediment-water interface. Second, the competitive behavior of OH^- and PO_4^{3-} may be considered as the second part. Third, the reverse reaction of Fe^{3+} hydrolysis is enhanced by a weakly acidic environment [18,19].

Released by excess Fe^{3+} from Fe-P can inhibit the growth of *C. vulgaris* and even exhibit a toxic

effect, which is unfavorable for the treatment of eutrophic wastewater [20,21]. Periodical algal blooms are induced by the release of P from P-containing sediments [22,23]. In addition, Fe^{3+} has a strong flocculation effect on *C. vulgaris* because the surface of *C. vulgaris* is negatively charged and evenly suspended in liquid [24,25]. Therefore, chain reactions involving biomass production, pH fluctuation, flocculation, co-precipitation, and removal of P can be triggered through the release dynamics of Fe-P. However, these processes have not been used to evaluate the advantages and disadvantages of *C. vulgaris*.

In realistic applications, phosphorus-contaminated water bodies are repaired by *C. vulgaris* in an Fe-P-rich environment [26-29]. To simulate actual environmental conditions, in this study, using different amounts of FeCl_3 and high content K_2HPO_4 to quantify Fe-P flocculated sediment, *C. vulgaris* was cultured in BG-11 medium (removed PO_4^{3-} and ferric ammonium) with Fe-P during a continuous three-cycle culture. The effects of $\text{Fe}^{3+}/\text{PO}_4^{3-}$ release from Fe-P on biomass, PO_4^{3-} removal, and flocculation were investigated. Finally, this study evaluated the environmental effects of *C. vulgaris* in Fe-P-rich sediment rivers.

Point 3: Line 17. The full name of *C. vulgaris* should be given.

Response 3: The full name of *C. vulgaris* (*Chlorella vulgaris*) has been modified in the abstract section (line 17).

Point4: Line 38. phosphorous? Pls check the similar mistakes throughout the manuscript.

Response 4: The word error has been corrected. It should be phosphorus. At the same time, other similar problems were checked and corrected.

Point 5: Line 42. *Chlorella vulgaris* (*C. vulgaris*)?

Response 5: *Chlorella vulgaris* is abbreviated as *C. vulgaris* or *C. v.* The former is used in this article. It is based on the results of many articles (Uniform writing of many references).

Point 6: Line 50. phosphate should be phosphate (PO_4^{3-}).

Response 6: “phosphate” is replaced by “ PO_4^{3-} ”.

Point 7: Line 59. biology?

Response 7: This word has been deleted.

Point 8: Line 66. but three general theories?

Response 8: This sentence has been replaced by a new one: There are several mechanisms that may be used to explain the release of Fe-P.

Point 9: Line 77. Re?

Response 9: This word (Re-release) has been modified. “Re-” has been deleted.

Point 10: Line 120. As shown in Table 1, all the spiked Fe^{3+} was transformed to Fe-P?

Response 10: The concentration of PO_4^{3-} in the medium is as high as 60mg/L. However, the group with the highest Fe^{3+} concentration is 25mg/L, so all the spiked Fe^{3+} was transformed to Fe-P during the experiment. If there is a little error, the change of the iron ion concentration in the solution shown in Figure 1 still has a reference degree according to the actual measurement. In addition, the error of the parallel sample shown in Figure 1 is also very small.

Point 11: Line 204. this stage can be regarded as the adsorption of *C. vulgaris*?

Response 11: Because Fe^{3+} in the solution are significantly reduced with the increase of flocculation efficiency in this stage. Simultaneously, the growth of biomass is weakly inhibited according to Figure 3b. The adsorption stage of microalgae is proved by the data of biomass and flocculation efficiency.

Point 12: Line 230. “The amount of released Fe^{3+} is the only factor that affects the *C. vulgaris* growth”, the algae only need Fe^{3+} to sustain their growth?

Response 12: Here is the author did not express clearly, further explanation has been added. This sentence means that other background conditions (such as pH temperature, light, BG-11, etc.) are the same. The amount of released Fe^{3+} becomes the dominant factor affecting the experimental process.

Point 13: Line 323. “To investigate the effect of Fe-P release on P removal”, removed from the solution or sediment? It is hard to understand the authors’ opinions.

Response 13: Here is the author did not express clearly. This sentence has been rephrased: To investigate the effect of Fe-P sediment release on P removal in solution.

Appendix F

Response to RSC Associate Editor Comments

We thank you very much for giving us an opportunity to revise our manuscript, those comments are all valuable and very helpful for revising and improving our paper, as well as the important guiding significance to our researches. We have studied comments carefully and have made correction which we hope meet with approval. Revised portion are marked in red in the paper.

Point 1: The research presented in this draft is original and of interest to RSOS audience, however a professional proofreading service is highly recommended as the language needs a thorough revision in order to communicate your research clearly. For example, no sentences are provided in Section 2.2, which makes it difficult to understand it. Additionally, the quality of the figures must be improved, especially to avoid repetition. Please read carefully each of the comments from the reviewers and address each of them.

Response 1: Thanks for your guidance. Thank you for your guidance, it is helpful to improve the quality of my article. This paper has been reorganized, in order to make the article more logical. Rewritten for more concise and coherence. At the same time, a new title (Effects of *Chlorella vulgaris* on P release from ferric phosphate sediment by consecutive cultivation) is created. In addition, grammar problems are corrected through professional platforms "Paperpal Preflight".

The text description is added in section 2.2.

The comments of the two reviewers have helped me a lot. I have read it carefully and made changes based on the reviewers' comments. Revised portion are marked in red in the paper.

Appendix G

Response to Reviewer 2 Comments

Point: Two question,

1) The authors claim that they have polished the grammar, but I still find lots of sloppy English. For example, "The metabolites of *C. vulgaris* during the growth process can promote an increase in pH"? "in" should be "of". Pls improve the manuscript carefully again.

2) In the first round of review, I recommend the authors to correct the mistakes of the use of "full names" and (or) "abbreviated names", which don't mean that you need to give the full name of abbreviated name in the bracket following their abbreviated name or full name. For the 1st use of a Latin name of certain organism, you should use the full name, while for the 2nd use you can use the abbreviated name.

Response:

1) Thanks for your guidance, a professional proofreading service is carried out through the Charlesworth Author Services Team. The language of this version hopes to be approved:

I have completed comprehensive editing to improve word choice, word agreement, sentence construction, punctuation, emphasis, and flow of the text overall. I have edited the manuscript following American stylistic and spelling conventions. I edit out more words than I add (without losing content), always trying to make the writing as succinct and direct as possible.

2) I fully understand what you are saying and amended such issues. Correct use of abbreviations in the clean version of the manuscript.

Thank you for your time and work

Kind regards,

Chen